# Unsupervised Learning for Quadratic Assignment

## Abstract

We introduce PLUME search, a data-driven framework that enhances search efficiency in combinatorial optimization through unsupervised learning. Unlike supervised or reinforcement learning, PLUME search learns directly from problem instances using a permutation-based loss with a non-autoregressive approach. We evaluate its performance on the quadratic assignment problem, a fundamental NP-hard problem that encompasses various combinatorial optimization problems. Experimental results demonstrate that PLUME search consistently improves solution quality. Furthermore, we study the generalization behavior and show that the learned model generalizes across different densities and sizes.

## 1 Introduction

Combinatorial Optimization (CO) represents a central challenge in computer science and operations research, targeting optimal solutions within a large search space. CO encompasses numerous practical applications, including transportation logistics, production scheduling, network design, and resource allocation. The computational complexity of CO problems is typically NP-hard, making exact methods intractable for large instances. Researchers have thus developed approximation algorithms, heuristics, and hybrid approaches that balance solution quality with computational feasibility. These methods include simulated annealing, genetic algorithms, tabu search, and various problem-specific heuristics that can generate high-quality solutions within reasonable time budget (Kirkpatrick et al., 1983; Holland, 1992; Johnson and McGeoch, 1997; Glover and Laguna, 1998; Gomes and Selman, 2001; Blum and Roli, 2003).

**Data-driven Combinatorial Optimization** Recently, data-driven methods have gained significant attention in addressing combinatorial optimization problems. Taking the Travelling Salesman Problem (TSP) as an example, researchers have explored both Supervised Learning (SL) and Reinforcement Learning (RL) methods. In SL, approaches such as pointer networks and graph neural networks attempt to learn mappings from problem instances to solutions by training on optimal or near-optimal tours (Joshi et al., 2019; Vinyals et al., 2015). These models learn to imitate optimal or near-optimal solutions, leading to significant computational expense when building the training dataset. RL approaches frame TSP as a sequential decision-making problem where an agent learns to construct tours in a Markov decision process framework. While RL models have shown promise in small instances, these methods face significant challenges when scaling to larger problems. Furthermore, the sparse rewards and the high variance during training make it difficult for RL agents to learn effective policies (Bello et al., 2016).

An alternative approach is Unsupervised Learning (UL), which avoids sequential decision making and does not require labelled data. In (Min et al., 2023), the authors propose a surrogate loss for the TSP objective by using a soft indicator matrix $\mathbb{T}$ to construct the Hamiltonian cycle, which is then optimized for minimum total distance. The $\mathbb{T}$ operator can be essentially interpreted as a soft permutation operator, as demonstrated in (Min and Gomes, 2023), which represents a rearrangement of nodes on the route $1 \rightarrow 2 \rightarrow 3 \rightarrow \cdots \rightarrow n \rightarrow 1$, with $n$ representing the number of cities.

The TSP loss used in UL can be formally expressed in matrix notation. Essentially, we optimize:

$$\mathcal{L}_{\text{TSP}} = \langle \mathbb{T}\mathbb{V}\mathbb{T}^{\top}, \mathbf{D}_{\text{TSP}} \rangle, \tag{1}$$

where $\mathbb{V}$ represents a Hamiltonian cycle matrix encoding the route $1 \rightarrow 2 \rightarrow \cdots \rightarrow n \rightarrow 1$, $\mathbb{T}$ is an approximation of a hard permutation matrix $\mathbf{P}$, and $\mathbf{D}_{\text{TSP}}$ is the distance matrix with self-loop

distances set to $\lambda$. $\mathbb{T}\mathbb{V}\mathbb{T}^\top$ is a heat map that represents the probability that each edge belongs to the optimal solution, which guides the subsequent search process.

Since permutation operators are ubiquitous across many CO problems, and the application to TSP demonstrates their effectiveness, here, we extend this approach to a broader class of problems. Specifically, we propose **P**ermutation-based **L**oss with **U**nsupervised **M**odels for **E**fficient search (**PLUME** search), an unsupervised data-driven heuristic framework that leverages permutation-based learning to solve CO problems.

**Quadratic Assignment Problem**   The Quadratic Assignment Problem (QAP), essentially a permutation optimization problem, is an NP-hard problem with numerous applications across facility layout, scheduling, and computing systems. For example, in facility layout, QAP finds the optimal permutation of facilities minimizing total interaction costs based on inter-facility flows and inter-location distances. QAP's applications also include manufacturing plant design, healthcare facility planning, VLSI circuit design, telecommunications network optimization, and resource scheduling (Koopmans and Beckmann, 1957; Lawler, 1963).

Formally, QAP asks to assign $n$ facilities to $n$ locations while minimizing the total cost, which depends on facility interactions and location distances. A flow matrix $\mathbf{F} \in \mathbb{R}^{n \times n}$ captures the interaction cost between facility $i$ and facility $j$. Each location is represented by a matrix $\mathbf{X} \in \mathbb{R}^{n \times 2}$, where each row contains the coordinates of a location. The distance matrix $\mathbf{D} \in \mathbb{R}^{n \times n}$ is computed using the Euclidean distance $\mathbf{D}_{ij} = \|\mathbf{X}_i - \mathbf{X}_j\|_2$.

The objective is to find a permutation matrix $\mathbf{P} \in \{0, 1\}^{n \times n}$ that maps facilities to locations while minimizing the total cost, given by $\min_{\sigma \in S_n} \sum_{i=1}^{n} \sum_{j=1}^{n} F_{ij} D_{\sigma(i), \sigma(j)}$, where $\sigma$ is a permutation of $\{1, \ldots, n\}$ defining the assignment and $S_n$ denotes the set of all $n \times n$ permutation operators. This cost function can be equivalently written in matrix form as:

$$\min_{\mathbf{P} \in S_n} \langle \mathbf{P}\mathbf{F}\mathbf{P}^\top, \mathbf{D} \rangle, \tag{2}$$

where $\mathbf{P}$ is the permutation matrix representing the assignment.

**Unsupervised Learning for QAP**   In this paper, we use PLUME search to solve QAP. Following the TSP fashion in (Min et al., 2023), we use neural networks (NNs) coupled with a Gumbel-Sinkhorn operator to construct a soft permutation matrix $\mathbb{T}$ that approximates a hard permutation matrix $\mathbf{P}$. We use the soft permutation $\mathbb{T}$ to guide the subsequent search process, as shown in Figure 1. PLUME search is a neural-guided heuristic that learns problem representations through NNs

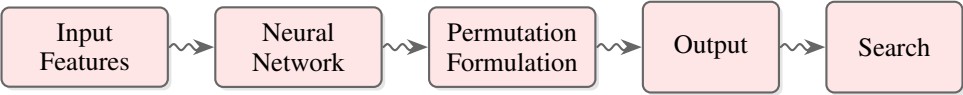

Figure 1: Overview of PLUME search framework: Input features are transformed through a neural network into a permutation formulation, then the output of the neural network guides the subsequent search process. This unified architecture allows PLUME to handle various combinatorial optimization problems by learning permutation operators.

to directly guide the search. By integrating learned representations, PLUME search leverages UL to enhance search performance. Here, for QAP, while TSP uses the heat map $\mathcal{H} = \mathbb{T}\mathbb{V}\mathbb{T}^\top$ as a soft matrix (heat map) to guide the search, we directly decode a hard permutation matrix $\mathbf{P}$ from $\mathbb{T}$. This permutation matrix $\mathbf{P}$ represents an alignment in QAP and serves as an initialization to guide the subsequent search.

## 2   Tabu Search

We adopt Tabu search as the backbone of our PLUME search framework. Tabu Search (TS) is a method introduced by Glover (Glover and Laguna, 1998) that enhances local search methods by employing memory structures to navigate the solution space effectively. Unlike traditional hill-climbing algorithms, TS allows non-improving moves to escape local optima and uses adaptive

memory to avoid cycling through previously visited solutions. The algorithm maintains a *tabu list* that prohibits certain moves, creating a dynamic balance between intensification and diversification strategies.

**Tabu Search for Quadratic Assignment**   TS has emerged as one of the most effective metaheuristics for addressing QAP instances (Taillard, 1991; Drezner, 2003; James et al., 2009). The algorithm navigates the solution space through strategic move evaluations and maintaining memory structures to prevent cycling and encourage diversification. For QAP, TS typically begins with a random permutation as the initial solution and employs a swap-based neighborhood structure where adjacent solutions are generated by exchanging the assignments of two facilities. A key component of TS is the tabu list, which records recently visited solutions or solution attributes to prevent immediate revisiting. In QAP implementations, the tabu list typically tracks recently swapped facility pairs, prohibiting their re-exchange for a specified number of iterations—the tabu tenure. This memory structure forces the search to explore new regions of the solution space even when immediate improvements are not available, helping the algorithm escape local optima. The performance depends mainly on three parameters: `neighbourhoodSize`, which controls the sampling density from the complete swap neighbourhood at each iteration; `evaluations`, which establishes the maximum computational budget as measured by objective function calculations; and `maxFails`, which implements an adaptive early termination criterion that halts the search after a predefined number of consecutive non-improving iterations. Together, these parameters balance exploration breadth against computational efficiency, ensuring both effective solution space coverage and predictable runtime performance (Glover, 1989; Blum and Roli, 2003; Battiti and Tecchiolli, 1994; Misevicius, 2005).

# 3   MODEL

We propose a permutation-equivariant neural architecture for the QAP. The network jointly encodes (i) pairwise *flows* among facilities and (ii) pairwise *distances* among locations, and fuses these with per-node positional features. Permutation equivariance is enforced by construction through row-wise symmetric pooling operators (sum, mean, max), shared multilayer perceptron (MLP) encoders for facility–facility and location–location interactions, and equivariant message passing updates that avoid in-place modifications to ensure stable gradient flow. As shown in Figure 2, our neural network

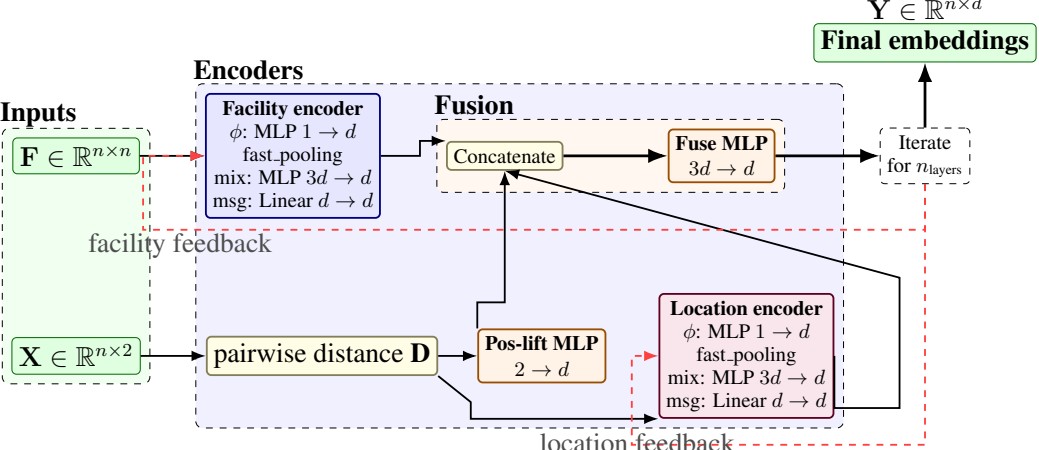

Figure 2: Neural network architecture for QAP. The model takes coordinates and flow matrix data and passes them into MLP encoders, the resulting features are then concatenated together. We then feed them into the main network and generate $\mathbf{Y}$ through an output layer.

consists of a facility encoder and a location encoder, each based on lightweight MLPs and pooled symmetric operators, a positional lifting module that embeds 2D coordinates into the hidden space, and a fusion block that integrates facility, location, and positional features across multiple layers.

By construction, our model guarantees permutation equivariance, in the sense that permuting rows and columns of the inputs permutes the learned embeddings in the same way.

Let $n$ denote the number of facilities/locations. The inputs of our model are 2D coordinates $\mathbf{X} \in \mathbb{R}^{n \times 2}$ and a (typically symmetric) flow matrix $\mathbf{F} \in \mathbb{R}^{n \times n}$. We precompute once per batch the Euclidean distance matrix $\mathbf{D} \in \mathbb{R}^{n \times n}$. $\mathbf{X}$ is embedded once by a 2-layer MLP $\phi_x : \mathbb{R}^2 \to \mathbb{R}^d$,

$$\mathbf{H}^{\text{pos}} = \phi_x(\mathbf{X}) \in \mathbb{R}^{n \times d}, \tag{3}$$

and reused across layers as fixed positional context.

**Encoders**   The encoders provide node-aligned embeddings of flows, distances, and coordinates that respect permutation equivariance. Each encoder lifts scalar pairwise inputs into $\mathbb{R}^d$, aggregates across neighbors using symmetric pooling (sum/mean/max), mixes the pooled summary with an MLP, and applies a row-stochastic message pass. This design ensures that both global context and local interactions are captured before embeddings are fused downstream.

Our flow encoder transforms entries of the flow matrix $\mathbf{F} \in \mathbb{R}^{n \times n}$ with a shared 2-layer MLP $\phi_f : \mathbb{R} \to \mathbb{R}^d$: $\mathcal{X}_{ij} = \phi_f(\mathbf{F}_{ij})$, for each node $i$, pooled statistics over its neighbors are computed:

$$\mathbf{s}_i^{\Sigma} = \sum_j \mathcal{X}_{ij}, \quad \mathbf{s}_i^{\mu} = \tfrac{1}{n}\mathbf{s}_i^{\Sigma}, \quad \mathbf{s}_i^{\max} = \max_j \mathcal{X}_{ij}. \tag{4}$$

These are concatenated into $\mathbf{h}_i^{\text{pool}} = [\mathbf{s}_i^{\Sigma} \parallel \mathbf{s}_i^{\mu} \parallel \mathbf{s}_i^{\max}] \in \mathbb{R}^{3d}$ and passed through a mixing MLP $\text{mix}_f : \mathbb{R}^{3d} \to \mathbb{R}^d$ to obtain $\tilde{\mathbf{h}}_i$. To propagate information along flow structure, we normalize $\mathbf{F}$ row-wise:

$$\mathbf{W}_F = \text{normalize}_1(\mathbf{F}), \tag{5}$$

and apply a linear message map $\text{msg}_f : \mathbb{R}^d \to \mathbb{R}^d$, giving the encoder output

$$\mathbf{H}^{\text{fac}} = \tilde{\mathbf{h}} + \mathbf{W}_F \, \text{msg}_f(\tilde{\mathbf{h}}) \in \mathbb{R}^{n \times d}. \tag{6}$$

Thus, each facility embedding combines local pooled features with weighted messages from other facilities, scaled by flow magnitude.

Our location encoder is structurally identical but uses distances. A shared 2-layer MLP $\phi_\ell : \mathbb{R} \to \mathbb{R}^d$ embeds distances into $\mathcal{G}_{ij} = \phi_\ell(D_{ij})$, producing pooled summaries $\mathbf{z}_i^{\text{pool}} \in \mathbb{R}^{3d}$. After mixing, we obtain $\tilde{\mathbf{z}} = \text{mix}_\ell(\mathbf{z}^{\text{pool}})$. To emphasize nearby nodes, we construct a kernel from the inverse distance:

$$\mathbf{W}_D = \text{normalize}_1\big((\mathbf{D} + \epsilon \mathbf{1})^{-1}\big), \qquad \epsilon = 10^{-3}, \tag{7}$$

and update embeddings via

$$\mathbf{H}^{\text{loc}} = \tilde{\mathbf{z}} + \mathbf{W}_D \, \text{msg}_\ell(\tilde{\mathbf{z}}) \in \mathbb{R}^{n \times d}. \tag{8}$$

This encourages each node to aggregate more strongly from spatially close locations, consistent with the QAP cost structure.

Finally, raw coordinates are embedded by a 2-layer position lift MLP $\phi_x : \mathbb{R}^2 \to \mathbb{R}^d$:

$$\mathbf{H}^{\text{pos}} = \phi_x(\mathbf{X}) \in \mathbb{R}^{n \times d}, \tag{9}$$

computed once and reused across layers. These positional embeddings act as fixed context that complements the dynamic flow and distance streams.

**Fusion Stack and State Update**   To enable effective message passing, we introduce a *Fusion Stack* combined with a *State Update* mechanism. The key idea is to iteratively refine node embeddings so that information propagates across the entire graph, thereby capturing multi-hop dependencies between facilities and locations. This is crucial for QAP, since the cost of assigning a facility to a location depends not only on its direct interaction with a single other facility but also on indirect chains of interactions involving the rest of the system.

At layer $\ell$, we construct node-aligned streams for facilities, locations, and positions, concatenate them, and fuse with a learned transformation:

$$\mathbf{U}^{(\ell)} = \big[\mathbf{H}^{\text{fac}} \parallel \mathbf{H}^{\text{loc}} \parallel \mathbf{H}^{\text{pos}}\big] \in \mathbb{R}^{n \times 3d}, \qquad \mathbf{H}^{(\ell)} = \text{fuse}(\mathbf{U}^{(\ell)}) \in \mathbb{R}^{n \times d}, \tag{10}$$

where fuse : $\mathbb{R}^{3d} \to \mathbb{R}^d$ is implemented as a 3-layer MLP. We then perform a state update by setting

$$\mathbf{H}^{\text{fac}} \leftarrow \mathbf{H}^{(\ell)}, \qquad \mathbf{H}^{\text{loc}} \leftarrow \mathbf{H}^{(\ell)}, \tag{11}$$

so that both facility and location streams share the updated node representation, while $\mathbf{H}^{\text{pos}}$ remains fixed as positional context. After $n_{\text{layers}}$ layers of Fusion and State Update, the network outputs embeddings

$$\mathbf{Y} = \mathbf{H}^{(n_{\text{layers}})} \in \mathbb{R}^{n \times d}. \tag{12}$$

An additional advantage of this design is that the weighting of message updates is directly aligned with the QAP objective. We apply row normalization to the flow matrix $\mathbf{F}$ and to the inverse distance matrix $(\mathbf{D} + \epsilon)^{-1}$, yielding row-stochastic kernels that act as attention weights. This ensures that messages are propagated preferentially along cost-critical interactions: strong flows between facilities and short distances between locations. As a result, the network allocates representational capacity to the most influential dependencies while still retaining global structural context through symmetric pooling. This balance between broad structural awareness and targeted local refinement enables the model to capture higher-order dependencies that are essential for achieving globally optimal assignments.

### 3.1 Building Soft Permutation $\mathbb{T}$

Our model first generates logits which are transformed by a scaled hyperbolic tangent activation:

$$\mathcal{F} = \alpha \tanh(\mathbf{Y}\mathbf{Y}^\top), \tag{13}$$

where $\alpha$ is a scaling parameter that controls the magnitude of the output. We then construct a soft permutation matrix $\mathbb{T}$ using the Gumbel-Sinkhorn operator:

$$\mathbb{T} = \text{GS}\left(\frac{\mathcal{F} + \gamma \times \text{Gumbel noise}}{\tau}, l\right), \tag{14}$$

where GS denotes the Gumbel-Sinkhorn operator that builds a continuous relaxation of a permutation matrix. Here, $\gamma$ controls the scale of the Gumbel noise which adds stochasticity to the process, $\tau$ is the temperature parameter that controls the sharpness of the relaxation (lower values approximate discrete permutations more closely), and $l$ is the number of Sinkhorn normalization iterations.

During inference, we obtain a discrete permutation matrix $\mathbf{P}$ by applying the Hungarian algorithm to the scaled logits: $\mathbf{P} = \text{Hungarian}(-\frac{\mathcal{F} + \gamma \times \text{Gumbel noise}}{\tau})$. We use the CUDA implementation of the batched linear assignment solver for the Hungarian operator from (Karpukhin et al., 2024).

### 3.2 Invariance Property of Permutation Representation

The soft permutation matrix $\mathbb{T}$ is constructed through Equations 13 and 14 to preserve permutation equivariance. Let $\Pi \in S_n$ represent a random permutation on the nodes, the distance and flow matrices transform as $\mathbf{D} = \Pi \mathbf{D}_0 \Pi^\top$ and $\mathbf{F} = \Pi \mathbf{F}_0 \Pi^\top$, where $\mathbf{D}_0$ and $\mathbf{F}_0$ are the distance matrix and flow matrix before this random permutation respectively. Now, let $\mathbf{Y}_0$ denote the initial output, given that the network output transforms under permutation $\Pi$ as $\mathbf{Y} = \Pi \mathbf{Y}_0$, we then have $\mathbb{T} = \Pi \mathbb{T}_0 \Pi^\top$. Consequently, the objective function $\langle \mathbb{T}\mathbf{F}\mathbb{T}^\top, \mathbf{D} \rangle$ remains invariant. To be specific, $\forall\, \Pi \in S_n$, we have:

$$\langle \mathbb{T}\mathbf{F}\mathbb{T}^\top, \mathbf{D} \rangle = \langle \Pi\mathbb{T}_0\Pi^\top\left(\Pi\mathbf{F}_0\Pi^\top\right)\left(\Pi\mathbb{T}_0\Pi^\top\right)^\top, \Pi\mathbf{D}_0\Pi^\top \rangle$$

$$= \langle \Pi\mathbb{T}_0 \underbrace{\Pi^\top\Pi}_{=I}\mathbf{F}_0\underbrace{\Pi^\top\Pi}_{=I}\mathbb{T}_0^\top\Pi^\top, \Pi\mathbf{D}_0\Pi^\top \rangle = \langle \Pi\mathbb{T}_0\mathbf{F}_0\mathbb{T}_0^\top\Pi^\top, \Pi\mathbf{D}_0\Pi^\top \rangle \tag{15}$$

$$= \langle \Pi\left(\mathbb{T}_0\mathbf{F}_0\mathbb{T}_0^\top\right)\Pi^\top, \Pi\mathbf{D}_0\Pi^\top \rangle = \langle \mathbb{T}_0\mathbf{F}_0\mathbb{T}_0^\top, \mathbf{D}_0 \rangle.$$

This invariance property guarantees consistent solutions for isomorphic problem instances. Overall, Equations 13 and 14 preserve permutation symmetry while enabling gradient-based optimization, with the additional benefit of allowing the model to naturally generalize across different problem sizes. This generalization capability arises because the soft permutation matrix $\mathbb{T} \in \mathbb{R}^{n \times n}$ always matches the input graph size, independent of the model's parameters. Consequently, our approach scales to problems of varying sizes without needing to modify the NN.

## 4 RESULTS

**Data Generation and Training**   We generate synthetic instances using an Erdős-Rényi (ER) graph model following (Tan and Mu, 2024). To build a QAP instance of size $n$, we generate a flow matrix $\mathbf{F} \in \mathbb{R}^{n \times n}$ and the location coordinates $\mathbf{X} = (\texttt{Uniform}(0, 1), \texttt{Uniform}(0, 1))$. We generate uniformly random weights in $[0, 1]$ for the upper triangular portion of $\mathbf{F}$, then mirror these values to create a symmetric matrix. We apply an ER graph mask with edge probability $p$ to control the sparsity of connections between facilities.

Formally, for each QAP instance $i$:

$$\mathbf{F}_{ij} = \begin{cases} \texttt{Uniform}(0, 1) & \text{if rand}() < p \text{ and } i \neq j \\ 0 & \text{otherwise} \end{cases}, \quad \forall i < j \tag{16}$$

$$\mathbf{F}_{ji} = \mathbf{F}_{ij}, \quad \forall i < j. \tag{17}$$

We build datasets with varying problem sizes $n \in \{100, 200\}$ and graph densities $p \in \{0.1, 0.2, \dots, 0.9\}$. For each configuration, we generate 30,000 instances for training, 5,000 for validation and 5,000 for test, respectively. We run experiments using an NVIDIA H100 GPU and an Intel Xeon Gold 6154 CPU. We optimize our NNs to minimize the QAP objective:

$$\langle \mathbb{T}\mathbf{F}\mathbb{T}^{\top}, \mathbf{D} \rangle, \tag{18}$$

with the model's hidden dimension set to either 128 or 256. The number of layers $n_{\text{layers}}$ is set to 3. For the Gumbel-Softmax operator used in Equation 13, we set $\tau = 3$ and $l = 100$. The noise scale $\gamma = 0.01$. The $\texttt{tanh}$ scale is set to $\alpha = 40$. We use the AdamW optimizer with a learning rate of $3 \times 10^{-5}$ and train for 300 epochs. We train our model on each problem size $n$ and each graph density $p$. After training the model, we then validate it and select the best parameters before testing. We implement a PLUME search in a straightforward way. As mentioned, the model outputs the soft permutation matrix $\mathbb{T}$, and we decode the hard permutation matrix $\mathbf{P}$ from $\mathbb{T}$. $\mathbf{P}$ corresponds to a learned assignment. We then start the tabu search for QAP using this learned assignment as the initial solution.

Table 1: Comparison between the average costs of UL predicted solutions and random solutions. The gap value indicates the average percentage improvement of the predicted solution over the random solution's cost. Inference time denotes the average total time (running NN inference+Hungarian) required to build the UL predicted solutions.

| | $n = 100$ | | | | $n = 200$ | | | |
|---|---|---|---|---|---|---|---|---|
| $p$ | UL | random | Gap (%) | Inference (ms) | UL | random | Gap (%) | Inference (ms) |
| 0.1 | 214.169 | 257.877 | 16.95 | 0.5747 | 913.790 | 1037.01 | 11.88 | 2.0885 |
| 0.2 | 457.821 | 515.381 | 11.17 | 0.5686 | 1908.48 | 2074.41 | 8.00 | 2.0589 |
| 0.3 | 709.034 | 773.459 | 8.33 | 0.5672 | 2917.46 | 3110.27 | 6.20 | 2.0049 |
| 0.4 | 960.346 | 1031.92 | 6.94 | 0.5673 | 3939.86 | 4144.96 | 4.95 | 1.9856 |
| 0.5 | 1214.74 | 1288.90 | 5.75 | 0.5633 | 4963.71 | 5184.32 | 4.26 | 1.9822 |
| 0.6 | 1472.90 | 1547.62 | 4.83 | 0.5588 | 5996.06 | 6219.82 | 3.60 | 1.9626 |
| 0.7 | 1729.11 | 1805.83 | 4.25 | 0.5572 | 7035.88 | 7257.90 | 3.06 | 1.9375 |
| 0.8 | 1989.98 | 2064.49 | 3.61 | 0.5535 | 8072.96 | 8298.41 | 2.72 | 1.8735 |
| 0.9 | 2251.18 | 2320.06 | 2.97 | 0.5488 | 9122.38 | 9336.99 | 2.30 | 1.9690 |
| Mean | 1222.14 | 1289.50 | 7.20 | 0.5621 | 4985.62 | 5184.90 | 5.22 | 1.9736 |

**Effectiveness of UL-Based Initialization**   Before diving into the PLUME search's final results, we first check whether the learned assignment improves the solution quality without any subsequent search. We directly compare the quality of solutions using learned assignments versus random assignments in the initialization stage.

Table 1 demonstrates the cost of using UL-predicted solutions compared to random initialization. We define the gap as: $1 - \frac{\langle \mathbf{P}\mathbf{F}\mathbf{P}^{\top}, \mathbf{D} \rangle}{\langle \mathbf{P}_{random}\mathbf{F}\mathbf{P}_{random}^{\top}, \mathbf{D} \rangle}$, where $\mathbf{P}$ is the learned assignment and $\mathbf{P}_{random}$ is a

random assignment. The gap percentage shows consistent improvement across all problem densities, with more significant gains observed in sparser problems. For $n = 100$ with density $p = 0.1$, the UL approach achieves a 16.95% improvement over random initialization, while for $n = 200$, it yields a 11.88% improvement. As problem density increases, the gap narrows but remains positive, indicating that our approach maintains its advantage even for denser problems.

Table 2: Performance comparison of selected tabu search configurations on QAP instances with $n = 100, 200$. $\text{TS}(\mu, \kappa, \omega)$ denotes the tabu search with `evaluations` $: \mu$, `neighbourhoodSize` $: \kappa$, and `maxFails` $: \omega$.

| | $n = 100$ | | | | $n = 200$ | | | |
| | $\text{TS}(1k, 25, 25)$ | | $\text{TS}(10k, 100, 100)$ | | $\text{TS}(1k, 25, 25)$ | | $\text{TS}(10k, 100, 100)$ | |
| $p$ | UL | random | UL | random | UL | random | UL | random |
|---|---|---|---|---|---|---|---|---|
| 0.1 | 186.255 | 198.931 | 164.509 | 167.250 | 855.216 | 918.224 | 792.157 | 816.504 |
| 0.2 | 417.235 | 434.907 | 386.651 | 390.903 | 1827.18 | 1913.14 | 1740.81 | 1773.71 |
| 0.3 | 659.526 | 679.353 | 623.167 | 627.494 | 2820.25 | 2921.98 | 2718.91 | 2758.39 |
| 0.4 | 905.508 | 928.427 | 865.668 | 871.310 | 3830.55 | 3937.86 | 3718.45 | 3758.61 |
| 0.5 | 1156.03 | 1179.51 | 1113.97 | 1119.23 | 4846.67 | 4965.70 | 4728.44 | 4776.09 |
| 0.6 | 1411.77 | 1435.01 | 1368.36 | 1373.03 | 5874.34 | 5994.81 | 5752.03 | 5800.11 |
| 0.7 | 1667.47 | 1692.78 | 1624.33 | 1630.24 | 6912.64 | 7032.42 | 6790.50 | 6836.88 |
| 0.8 | 1928.97 | 1953.37 | 1886.45 | 1892.27 | 7950.10 | 8076.76 | 7829.94 | 7884.92 |
| 0.9 | 2191.91 | 2213.92 | 2151.28 | 2155.33 | 9003.56 | 9125.34 | 8888.56 | 8941.46 |
| Average | 1169.41 | 1190.69 | 1131.60 | 1136.34 | 4880.06 | 4987.36 | 4773.31 | 4816.30 |

**PLUME Search for QAP** We then run tabu search using different initializations. Tables 2 shows the performance of PLUME tabu search compared with tabu search with random initialization. Our experimental results demonstrate that our UL-based method effectively solves QAPs. The solutions consistently outperform random initialization across all problem sizes and density parameters. Specifically, we vary the `evaluations` $: \mu$, `neighbourhoodSize` $: \kappa$, `maxFails` $: \omega$ and test PLUME search. We show that our UL-based initialization consistently outperforms random initialization within each parameter set, and the solution quality improves with increased `evaluations` $\mu$.

Table 3: Average time comparison of selected tabu search configurations on QAP instances with $n = 100$ and $n = 200$, in ms ($\times 10^{-3}$ s). $\text{TS}(\mu, \kappa, \omega)$ denotes the tabu search with `evaluations` $: \mu$, `neighbourhoodSize` $: \kappa$, and `maxFails` $: \omega$.

| | $n = 100$ | | | | $n = 200$ | | | |
| | $\text{TS}(1,000, 25, 25)$ | | $\text{TS}(10,000, 100, 100)$ | | $\text{TS}(1,000, 25, 25)$ | | $\text{TS}(10,000, 100, 100)$ | |
| $p$ | UL | random | UL | random | UL | random | UL | random |
|---|---|---|---|---|---|---|---|---|
| 0.1 | 1.55014 | 1.47786 | 6.89681 | 6.40729 | 2.42600 | 2.29224 | 11.8025 | 11.8930 |
| 0.2 | 1.40809 | 1.72910 | 6.19193 | 6.92497 | 2.29030 | 2.33664 | 11.9798 | 11.8780 |
| 0.3 | 2.23220 | 2.30824 | 6.34255 | 6.50180 | 2.34508 | 2.32658 | 11.9024 | 11.7376 |
| 0.4 | 2.72354 | 2.91837 | 6.73373 | 6.90415 | 2.18291 | 2.57973 | 12.0805 | 11.9279 |
| 0.5 | 2.57762 | 1.60781 | 6.87188 | 6.36137 | 2.34563 | 2.22272 | 12.0350 | 11.9095 |
| 0.6 | 1.90425 | 1.71858 | 6.63121 | 6.72185 | 2.54630 | 2.29580 | 12.0519 | 11.6976 |
| 0.7 | 1.60661 | 1.57086 | 6.60751 | 6.77780 | 2.27941 | 2.27283 | 12.1370 | 11.8553 |
| 0.8 | 1.91665 | 2.06804 | 6.81585 | 6.88984 | 2.35310 | 2.39341 | 11.6703 | 11.8593 |
| 0.9 | 2.42608 | 2.11719 | 6.28867 | 6.47949 | 2.18062 | 2.51180 | 11.6522 | 11.5438 |
| Mean | 2.03835 | 1.94623 | 6.59779 | 6.66317 | 2.32771 | 2.35908 | 11.92351 | 11.81135 |

The performance gap is most pronounced in sparse problems (low $p$ values). With parameters $\mu = 1,000$, $\kappa = 25$, and $\omega = 25$, PLUME search yields a 6.37% improvement at $p = 0.1$, $n = 100$, and a

6.86% improvement at $p = 0.1$, $n = 200$. This advantage diminishes as problem density increases, indicating that our models are particularly effective at capturing structural patterns in sparse settings.

The relative improvement from using PLUME search is more significant for larger problem sizes, with $n = 200$ showing consistently higher percentage improvements than $n = 100$ at comparable densities. This trend holds for tabu search with different parameters. For example, with parameters $\mu = 1,000$, $\kappa = 25$, and $\omega = 25$, the tabu search with random initialization achieves 1190.69 on average for $n = 100$, while PLUME search achieves 1169.41, yielding a 1.79% improvement. At $n = 200$, PLUME search achieves 4880.06 on average while random initialized tabu search achieves 4987.36, resulting in a more substantial 2.15% improvement. The most intensive tabu search configuration, TS$(10,000, 100, 100)$, provides the most comprehensive exploration of the solution space and thus yields the best solution quality across all initialization methods. At this parameter setting, PLUME search further improves solution quality. For graphs with $n = 100$, PLUME search achieves an average solution cost of 1131.60 compared to 1136.34 for random initialization, yielding a 0.42% improvement. While this may appear modest, it is important to note that as search parameters increase, the final solutions converge closer to optimality, leaving less room for improvement. The fact that PLUME search maintains its advantage even in this setting underscores the quality of its initialization. For larger graphs with $n = 200$ vertices, the benefit becomes more pronounced. PLUME search achieves an average cost of 4773.31 compared to 4816.30 for random initialization, yielding a 0.89% improvement. This increasing advantage with problem size suggests that PLUME search scales well to larger problems.

An important observation emerges when analyzing initialization performance at larger problem scales. For $n = 200$, the UL-based initialization produces solutions of comparable quality to those obtained by a lightweight tabu search configuration, TS$(1,000, 25, 25)$. This finding highlights that the UL model does not merely provide a heuristic shortcut, but rather learns to internalize and exploit the underlying problem structure. Consequently, the UL-based initialization can be interpreted as a learned search mechanism in its own right, capable of directly yielding high-quality solutions without requiring extensive local exploration.

Tables 3 shows the execution time comparison (TS runtime) between random and UL-based initialization approaches for tabu search. The inference time for generating UL-predicted solutions is 0.56 ms for $n = 100$ and 1.98 ms for $n = 200$ (see Table 1). As discussed, the UL-based initialization on $n = 200$ achieves competitive performance with respect to TS with parameters $\mu = 1,000$, $\kappa = 25$, and $\omega = 25$ (4985.62 vs. 4987.36), while the TS runtime is 2.35 ms. It should be noted that our UL inference is performed on GPU, whereas tabu search is executed on CPU; therefore, the runtimes are not directly comparable. Nevertheless, the shorter inference cost highlights that UL is an efficient and effective method for initialization.

Table 4: Cross-size generalization: Models trained and tested on different problem sizes (both with $p = 0.7$). Random Initialization shows cost from random assignments; UL-based Initialization shows cost from UL model. TS$(\mu, \kappa, \omega)$ shows costs after running tabu search from random initialization. PLUME TS$(\mu, \kappa, \omega)$ shows costs after running tabu search from UL-predicted initialization.

| Train Size | Test Size | Random Initialization | UL-based Initialization | TS $(1k, 25, 25)$ | PLUME TS $(1k, 25, 25)$ | TS $(10k, 100, 100)$ | PLUME TS $(10k, 100, 100)$ |
|---|---|---|---|---|---|---|---|
| 200 | 100 | 1805.83 | 1754.69 | 1692.78 | 1674.92 | 1630.24 | 1625.90 |
| 100 | 200 | 7257.90 | 7081.93 | 7023.566 | 6934.05 | 6836.88 | 6798.40 |

**Compared with Other Data-driven Methods** We compare with recent work by (Tan and Mu, 2024), where the authors use RL and test only on QAP instances up to size 100. We run their model on $n = 100$ and $p = 0.7$, using their model, we observe an average cost of 1644.37 with an average time of $\approx 150$ ms, our PLUME TS$(10,000, 100, 100)$ achieves better performance with a cost of 1624.33. Notably, our methods are substantially faster, with average time costs within 10 ms. It should be noted that although (Tan and Mu, 2024) also uses tabu search as a benchmark; they employ a Python implementation, which is not as computationally efficient as our C++ implementation of tabu search. We didn't fine-tune the RL method and fine-tuning may yield better performance.

However, the gaps are so dramatically large that even with optimization, the RL approach would remain substantially inferior.

## 5 GENERALIZATION

Table 5: Generalization across selected densities for models trained on $n = 100$, $p = 0.7$ and $n = 200$, $p = 0.7$. Random initialization shows costs using random initial assignments. UL-based initialization shows costs using assignments predicted by our neural network. TS$(\mu, \kappa, \omega)$ shows costs after running tabu search from random initialization. PLUME TS$(\mu, \kappa, \omega)$ shows costs after running tabu search from UL-predicted initialization.

|  |  | Random | | TS$(1,000, 25, 25)$ | | TS$(10,000, 100, 100)$ | |
|---|---|---|---|---|---|---|---|
| $n$ | $p$ | Random | UL-based | Random | PLUME | Random | PLUME |
| 100 | 0.6 | 1547.62 | 1500.26 | 1435.01 | 1420.25 | 1373.03 | 1370.25 |
| 100 | 0.8 | 2064.49 | 2058.36 | 1953.37 | 1950.79 | 1892.27 | 1891.46 |
| 200 | 0.6 | 6219.82 | 6205.04 | 5994.98 | 5984.02 | 5800.17 | 5795.99 |
| 200 | 0.8 | 8298.41 | 8283.00 | 8076.76 | 8060.64 | 7884.66 | 7874.34 |

**Cross-density Generalization** We further study how the model generalizes across different densities, as shown in Table 5. Using models trained on a specific edge density ($p = 0.7$), we test the model's performance on nearby densities ($p = 0.6$ and $p = 0.8$). Our results suggest the model captures transferable structural patterns that work best within a reasonable proximity to its training conditions. This effect is more pronounced with TS$(1,000, 25, 25)$ compared to configurations with more extensive search parameters. For $n = 100$ and $p = 0.6$, the UL-based initialization achieves a $3.06\%$ improvement over random initialization. When applying TS$(1,000, 25, 25)$, the performance gap between PLUME and random initialization is reduced to $1.03\%$, and with the stronger configuration TS$(10,000, 100, 100)$, the gap further narrows to $0.20\%$.

**Cross-size Generalization** Our model naturally generalizes across problem sizes due to its permutation-equivariant design, where the soft permutation matrix $\mathbb{T} \in \mathbb{R}^{n \times n}$ automatically adapts to match input dimensions. Experiments show that a model trained on $n = 100$ effectively generalizes to $n = 200$ problems and vice versa, while consistently outperforming random initialization, as shown in Table 4. For instance, when a model trained on $n = 200$ is applied to $n = 100$ problems, UL initialization achieves a solution cost of 1754.69 compared to 1805.83 for random initialization, and when used with TS$(1,000, 25, 25)$, PLUME TS reaches 1674.92 versus 1692.78 for standard TS with random initialization.

## 6 CONCLUSION

In this paper, we propose PLUME search, a framework to enhance combinatorial optimization through unsupervised learning. By leveraging a permutation-based loss, we demonstrate that neural networks can effectively learn the quadratic assignment problem directly from instances, rather than relying on supervised or reinforcement learning. Our experimental results indicate that UL can generate high-quality initial solutions that significantly outperform random initialization, and these improved starting points consistently lead to superior final solutions after tabu search, while also exhibiting strong generalization across varying problem densities and sizes.

A key insight is that our method does not merely act as an initializer, but rather learns to capture and exploit the intrinsic problem structure. In this sense, our unsupervised method can itself be regarded as a form of search, providing solutions of comparable quality to conventional heuristics after substantial exploration. PLUME search therefore takes a different path from traditional heuristics, offering a complementary paradigm that integrates seamlessly with existing frameworks rather than competing with them.

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
