# OpenReview forum: "Unsupervised Learning for Quadratic Assignment"
_ICLR.cc/2026/Conference — ICLR 2026 Conference Withdrawn Submission_

### Official Review · Reviewer_hPaN · 2025-10-16

**Soundness:** 3
**Presentation:** 3
**Contribution:** 3
**Rating:** 6
**Confidence:** 4

**Summary:**

In this paper, the authors propose an unsupervised learning model to optimise the Quadratic Assignment Problem. The authors propose using permutation-based loss functions that allow the characteristics of the input instances to be shared with the loss, enabling solutions to be constructed in a non-autoregressive manner. The obtained solution by the UL model is provided as starting solution to a Tabu Search algorithm.

**Strengths:**

S1. In combinatorial optimisation, it is well known that QAP is one of the most difficult problems. Not only that, but unlike others such as TSP, VRP, and LOP, it is not possible to represent it naturally in graph form so that it can be fed into a GNN. This means that for many years, QAP has not been on the list of papers to be optimised in the field of NCO. However, this paper addresses the challenge very satisfactorily and raises new ideas. I can only congratulate the authors on this success. I have been waiting a long time for someone to write this paper :-)

S2. The paper is in general well written and easy to read.

**Weaknesses:**

The paper has a number of issues that the authors should address:
W1. The unsupervised setting in combinatorial optimisation is not that common, so I think a clearer explanation is needed in the introduction and model part.
W2. One of the most interesting parts of the paper is the idea of permutation-based losses, but it is not entirely clear how the equations are computed. I would have liked to see a section 3.3. on this.
W3. The related work section needs improvement:
- There is no adequate justification for why the authors adopted a UL setting, when RL has been the norm for the last eight years.
- There are previous studies, albeit less efficient, that have addressed QAP (in addition to the 2024 study), and I believe the authors should include them.
W4. Considering that the authors adopt algebraic notation to represent the QAP, it is somewhat curious that they then refer to soft and hard permutation matrices, rather than doubly stochastic matrices or permutation matrices. In fact, in the metaheuristic field, QAP has been worked on using DSMs in the EDAs framework (see Santucci et al. 2024, On the use of the Doubly Stochastic Matrix models for the Quadratic Assignment Problem).
W5. The present work could have a huge impact if they carry out an exhaustive comparison with (Tan and Mu) and with the best metaheuristic for QAP (they could use DSM-EDA by Santucci et al.), or any of those reported in QAPlib (RoTS, etc.). As it is, the only thing the authors show is the possibility of doing optimization on the QAP with UL, but not its competitiveness (this also would be interesting, although not mandatory for publishing the present work).
W6. It is common to use u.a.r. instances in ML combinatorial optimisation, but from a realistic point of view, the instances generated are very different from what happens in reality. Perhaps the authors could consider QAPlib-style instances.

**Questions:**

Any progress in the weaknesses noted above would be positive for the paper, and I would reconsider the punctuation. In addition, I have some questions related:

Q1. Similar structures in heuristics field has been observed previously. The so-called GRASP is a concatenation of a constructive stochastic algorithm and a local search-based methods (like TabuSearch). Is this also applicable here?

Q2. How is the training process? which is the parametrisation? How do you describe it? (related to W2)

Q3. Why UL is more appropriate than RL for the QAP? Motivate properly the answer. (related to W3.1)

---

### Official Review · Reviewer_qmbg · 2025-10-25

**Soundness:** 2
**Presentation:** 3
**Contribution:** 1
**Rating:** 2
**Confidence:** 5

**Summary:**

The paper proposes **PLUME Search** (Permutation-based Loss with Unsupervised Models for Efficient Search), a **data-driven unsupervised learning framework** to improve **search efficiency in combinatorial optimization** (CO), specifically applied to the **Quadratic Assignment Problem (QAP)**. Unlike supervised or reinforcement learning approaches that rely on labeled data or sequential decision processes, PLUME directly learns from raw problem instances using a **permutation-based loss** and a **non-autoregressive neural architecture**. It provides learned heuristic initialization for classical solvers such as **Tabu Search**, improving both efficiency and solution quality.

**Strengths:**

- **\[S1] Important and general problem.** The paper tackles the **Quadratic Assignment Problem (QAP)**, a fundamental and notoriously hard combinatorial optimization problem that generalizes many practical tasks (e.g., layout design, scheduling, circuit placement). Extending data-driven optimization methods to QAP is both **technically challenging** and **practically significant**, since QAP captures a wide range of real-world assignment and matching scenarios.

**\[S2] Data-driven unsupervised approach is promising.** The proposed **unsupervised learning framework (PLUME Search)** eliminates the need for labeled solutions or costly reinforcement learning setups. By learning directly from raw problem instances, the model can serve as a **neural heuristic** or **learned search initializer**, making it a scalable and general alternative to traditional hand-designed heuristics.

**Weaknesses:**

See "Questions" below.

**Questions:**

- **\[Q1] On novelty over Min et al. (2023).** The core idea—unsupervised learning of soft permutation matrices via the Gumbel–Sinkhorn operator—appears conceptually similar to _Min et al. 2023._ Could the authors clarify what key methodological innovations distinguish PLUME Search from this prior work beyond applying it to the QAP setting (e.g., new theoretical insight, new training objective, or substantially different model behavior)?

- **\[Q2] On disentangling learning vs. post-processing contributions.** paper integrates unsupervised learning with a strong post-hoc Tabu Search. However, prior work (e.g., _Xia, Yifan, et al. "Position: Rethinking Post-hoc Search-based Neural Approaches for Solving Large-scale Traveling Salesman Problems." ICML 2024_; _Bu, Fanchen, and Kijung Shin. "On Training-Test (Mis)alignment in Unsupervised Combinatorial Optimization: Observation, Empirical Exploration, and Analysis." arXiv:2506.16732, 2025_) highlights that excessive reliance on search can obscure the learning contribution. Could the authors provide controlled experiments or ablation results that isolate how much of the observed performance gain arises from the learned model versus from Tabu Search itself? For example, how does the model perform under weaker or no post-processing, and how does it work when we apply the same post-processing to random initializations?

- **\[Q3] On alternative permutation-learning formulations.** The paper focuses exclusively on the Gumbel–Sinkhorn operator for differentiable permutation learning. Yet several recent works have proposed alternative or improved formulations, such as: _Dröge, Hannah, et al. "Kissing to Find a Match: Efficient Low-rank Permutation Representation." NeurIPS 2023._ and _Nerem, Robert R., et al. "Differentiable Extensions with Rounding Guarantees for Combinatorial Optimization over Permutations." NeurIPS 2024._ Could the authors discuss how their method compares theoretically or empirically to these alternatives?

---

### Official Review · Reviewer_qLh7 · 2025-10-31

**Soundness:** 3
**Presentation:** 1
**Contribution:** 2
**Rating:** 2
**Confidence:** 3

**Summary:**

This paper focuses on solving the NP-hard Quadratic Assignment Problem (QAP) in combinatorial optimization through unsupervised learning. The proposed PLUME search framework features key innovations: it adopts a permutation-based loss and non-autoregressive approach, leveraging a permutation-equivariant neural architecture with facility and location encoders to generate soft permutation matrices via the Gumbel-Sinkhorn operator, which are then decoded into hard permutations to initialize tabu search. Theoretically, the framework guarantees permutation equivariance and invariance of the QAP objective function, enabling natural generalization across different problem sizes. Experimentally, PLUME search consistently outperforms random initialization across various QAP problem sizes (100, 200) and densities, achieving improvement in initialization quality and maintaining advantages in tabu search results; it also exhibits strong cross-density and cross-size generalization, and outperforms RL-based methods in both solution quality and efficiency.

**Strengths:**

1. The model design presented in this paper is generally sound and comprehensive, covering the key components required for addressing the research problem.
2. The ablation studies and sensitivity analyses are thorough, providing strong support for the proposed method’s effectiveness, though, the work lacks comparisons with recent state-of-the-art (SOTA) baselines, which limits the full validation of its advantages.

**Weaknesses:**

1. The research motivation is not sufficiently solid. For instance, the paper claims that the studied problem leads to "significant computational expense when building the training dataset." Yet, taking the Traveling Salesman Problem (TSP) as a reference, heuristic algorithms like LKH (Lin-Kernighan-Helsgaun) can generate labels efficiently. For the Facility Location Problem  focused on in this paper, it would be better to more comprehensively justify the ability of unsupervised-based approach compared to supervised.

2. The paper lacks comparisons with existing works that also adopt unsupervised learning or Gumble-Sinkhorn-based for combinatorial optimization problems—especially for the Quadratic Assignment Problem (QAP), such as the studies cited in [1, 2].

3. A critical issue is that the baselines used for comparison are overly weak. In most experiments, only a random baseline is employed, which fails to demonstrate the proposed method’s competitiveness against meaningful benchmarks and greatly undermines the persuasiveness of the experimental results. Additionally, supplementing comparisons with representative supervised or reinforcement learning (RL)-based solvers would help contextualize the performance of the proposed unsupervised approach and highlight its relative merits.

4. The notation in Figure 2 is ambiguous, making it difficult to fully understand the model design purely from figure. For example, the transformation "3d→d" is not clearly explained, as the figure and its caption provide insufficient context. Furthermore, Equation 3 and Equation 9 appear to be duplicated, requiring clarification or correction.


[1] Wang, Runzhong, et al. "Unsupervised learning of graph matching with mixture of modes via discrepancy minimization." IEEE Transactions on Pattern Analysis and Machine Intelligence 45.8 (2023): 10500-10518.

[2] Wang, Runzhong, et al. "Towards one-shot neural combinatorial solvers: Theoretical and empirical notes on the cardinality-constrained case." The Eleventh International Conference on Learning Representations. 2022.

**Questions:**

1. What is rational of design of equation 3? Does the positional lifting model captures the pairwise positional relationship?
2. Does the design of flow encoder in equation 4 seems like a GNN with fully-connnected graph ?
3. What is the specific implementation of "msg" operation?

---

### Official Review · Reviewer_rxTq · 2025-10-31

**Soundness:** 3
**Presentation:** 2
**Contribution:** 2
**Rating:** 2
**Confidence:** 4

**Summary:**

This paper presents PLUME, an unsupervised learning framework for the Quadratic Assignment Problem (QAP). The approach is permutation-invariant and is trained via unsupervised learning, thereby avoiding the need to compute potentially expensive labels.  The model employs a permutation-equivariant encoder that embeds the flow and distance matrices symmetrically, ensuring invariance to label permutations.  Computationally, the unsupervised learning method demonstrates efficacy over random solutions.

**Strengths:**

- **Motivation**: The unsupervised learning approach is well motivated and sound, especially given that collecting labels for supervised learning is costly, and RL approaches tend to be challenging to train.
- **Results**: Overall, the reported results demonstrate that the unsupervised approach achieves relatively good quality solutions and is effective as a seed in Tabu search.  There are still some weaknesses in the results that I highlight below.

**Weaknesses:**

- **Benchmarks**: The authors currently evaluate their approach on randomly generated QAP instances with $n=100,200$.  However, there is no discussion on how the difficulty of these instances relates to the literature, e.g., QAPLib (some of which have generation schemes making them usable in a data-driven setting).  Given that the primary focus of this paper is on QAP, a more comprehensive evaluation of standard benchmark instances is needed.
- **Baselines**: The baselines are relatively weak, with Table 1 comparing random solutions, and Tables 2 and 3 reporting solution quality compared to Tabu search run with random initialization, given the offline time required to train the model and the inference time, comparing against more robust seeding/initialization should be done.
- **Results**: The authors report averaged results over the instances of each size.  However, this doesn't convey the statistical significance of the results; some distributional information, e.g., box plots, needs to be reported to compare with the best-known solutions.  In addition, the authors report inference time, but do not report training time or loss curves, which are absolutely needed. The authors do not explain how the training parameters were selected.  There should be an ablation on these.
- **Clarity on Contributions**: Given that the approach shares similarity with [1], a more precise delineation of the contributions of this work is needed.  This paper would benefit in terms of readability from a Contributions section.

## References
- [1] Yimeng Min and Carla Gomes. Unsupervised learning permutations for tsp using gumbel-sinkhorn
operator. In NeurIPS 2023 Workshop Optimal Transport and Machine Learning, 2023

**Questions:**

- How long does the unsupervised approach take to train?
- How much training data is required?
- How were the parameters selected, and is the training sensitive to hyperparameters?
- Can the approach be generalized to assignment problems where the number of facilities and locations differ?
- Do the authors think that this approach can be generalized more broadly to other classes of CO problems?

---

### Note · Authors · 2025-11-19

I have read and agree with the venue's withdrawal policy on behalf of myself and my co-authors.